# Household Food Insecurity During the COVID-19 Pandemic Between Slum and Non-Slum Areas in Kinshasa, DR Congo: A Cross-Sectional Study

**DOI:** 10.3390/foods13223657

**Published:** 2024-11-17

**Authors:** Pierre Z. Akilimali, Benito M. Kazenza, Francis K. Kabasubabo, Landry M. Egbende, Dynah M. Kayembe, Nguyen Toan Tran, Désiré K. Mashinda

**Affiliations:** 1Patrick Kayembe Research Center, Kinshasa School of Public Health, University of Kinshasa, Kinshasa P.O. Box 11850, Democratic Republic of the Congo; fkabasu13@gmail.com (F.K.K.); dirchkayembe@gmail.com (D.M.K.); 2Department of Nutrition, Kinshasa School of Public Health, University of Kinshasa, Kinshasa P.O. Box 11850, Democratic Republic of the Congo; benito.kazenza@unikin.ac.cd (B.M.K.); landry.egbende@unikin.ac.cd (L.M.E.); 3Australian Centre for Public and Population Health Research, Faculty of Health, University of Technology Sydney, P.O. Box 123, Sydney, NSW 2007, Australia; nguyentoan.tran@uts.edu.au; 4Faculty of Medicine, University of Geneva, Rue Michel-Servet 1, 1206 Geneva, Switzerland; 5Department of Biostatistics and Epidemiology, Kinshasa School of Public Health, University of Kinshasa, Kinshasa P.O. Box 11850, Democratic Republic of the Congo; desire.mashinda@unikin.ac.cd

**Keywords:** slum areas, food security, Kinshasa, post-pandemic recovery

## Abstract

Introduction: Food insecurity is a vital issue, especially in places such as Kinshasa. Additionally, food insecurity has been worsened by the COVID-19 pandemic, particularly in low- and middle-income countries. Thus, this study examined food insecurity in Kinshasa after the peak of the pandemic to understand the impact of post-pandemic recovery efforts as well as the heterogeneity of this problem according to the residence of respondent (slum vs. non-slum areas). Methods: Grounded in the four key dimensions of food security (availability, access, vulnerability, and utilization), this cross-sectional study was conducted in Kinshasa with a representative sample of 2170 households selected from 62 enumeration areas. We used a questionnaire to interview participants about their food situations. Interviews were conducted with the head of each household or their designated representative by 150 master’s students using tablets powered by the SurveyCTO application. Household food security status was evaluated using the Household Food Insecurity Access Scale. A logistic regression model was developed to assess household risk factors associated with food insecurity. Results: Most people we talked to were over 40 years old, and many lived in households with fewer than six people. About a third of the households were overcrowded. The prevalence of food insecurity was 76.5% (95%CI: 74.6–78.3). Factors associated with food insecurity included being a household head aged over 50 years, insufficient living space, lower socioeconomic status, and residing in slum areas (AOR: 1.38; 95% CI: 1.06–1.79). Conclusions: Vulnerable groups, such as slum residents, older adults, and informal workers are more likely to be affected by food insecurity. Addressing these challenges requires the government to develop targeted strategies that bolster resilience and mitigate household vulnerability during crises.

## 1. Introduction

Food security is achieved “when all people, at all times, have physical, social, and economic access to sufficient, safe, and nutritious food that meets their dietary needs and food preferences for an active and healthy life” [1,2,3]. This broad definition emphasizes four distinct dimensions of food security: the availability, accessibility, and utilization of food, in addition to the stability of each of these factors, which refers to the ability to withstand shocks to the broader food system [1,4]. Food insecurity occurs when at least one of these need domains is not met, during which the experience at the household level may be temporary or longer [5,6,7]. Access to adequate food is a core social determinant of health, and food insecurity is related to poor nutritional intake and higher mortality rates [2,6]. Even temporary reductions in food security can affect long-term health and cause a loss of human capital, from which it can take years to recover [8].

The recent COVID-19 pandemic increased the level of food insecurity worldwide, and low- and middle-income countries have been most affected [8]. More than 50% of households experienced food insecurity during the COVID-19 pandemic [2,6,7,8]; at the household level, it remains a major issue in many developing countries, particularly those in Africa. Although food insecurity remains high in low- and middle-income countries, many factors, such as poverty, exacerbate it. Food insecurity is also a significant risk factor for non-adherence to treatment, such as antiretroviral therapy among HIV-infected individuals [9]. In addition, factors such as living situation, low income, lack of livestock, high household size, and psychological situation (anxiety and depression) are principal aspects associated with food insecurity [2,5,8,10].

Previous studies on food security have been conducted in the Democratic Republic of the Congo (DRC) over the past decade, but they have often been limited in scope, focused on specific demographic groups or conducted on a relatively small scale. Furthermore, these studies predate the COVID-19 pandemic and are often characterized by descriptive rather than analytical approaches. In the first half of 2020, Performance Monitoring for Action reported a 40% prevalence of food insecurity in the city of Kinshasa [11]. However, not all urban residents uniformly experience food insecurity, and marginalized cities, commonly referred to as “slums”, represent the most significant examples of urban poverty in developing nations and are often the most impacted. Consequently, there is a pressing need to comprehensively assess food security on a broader scale, especially following the peak of the COVID-19 pandemic, and to delve into the factors contributing to it.

Slum areas, often marked by hazardous, unhealthy, and overcrowded housing with limited access to basic infrastructure, disproportionately expose residents to health issues compared to those in non-slum areas [12]. The rapid urbanization in developing countries, particularly in megacities, underscores the need for targeted health interventions, especially during crises [12]. Urbanization not only intrudes upon agricultural lands—diminishing local food production capabilities—but also leads to increased reliance on food imports, making urban regions vulnerable to price volatility and supply disruptions. This is further compounded by wealth disparity, with low-income households facing heightened vulnerability to food insecurity due to higher food expenditure burdens and price sensitivity [13]. Urban residents, particularly those in informal settlements, may struggle with limited access to affordable and healthy food, sanitation, clean water, and adequate food storage. These factors, intensified during events such as the COVID-19 pandemic, underscore the importance of comparing food insecurity across urban demographics, such as slum versus non-slum communities, to better understand the resilience of urban food systems.

In the DRC, urbanization and informal economies are increasingly central as formal employment remains scarce, with only 2.5% of the workforce employed in the formal sector, while the informal sector contributes around 42% of GDP [14]. The COVID-19 pandemic and associated lockdowns particularly impacted the informal sector, highlighting the need to understand food insecurity in the post-pandemic context. Assessing food insecurity in both slum and non-slum areas can reveal unique vulnerabilities and uncover at-risk groups, informing public health policies aimed at increasing resilience in the face of future crises. Therefore, this study aims to provide insights into household-level food insecurity in Kinshasa, examining its variation between slum and non-slum areas, especially among young women, and identifying contributing factors to guide effective interventions.

The DRC is the fourth most populous country in Africa [15]. Kinshasa, the capital of the DRC, is classified as one of the world’s “megacities”. In 2022, the metro area population of Kinshasa comprised 15,628,000 individuals [16]. Subsequently, Kinshasa ranks as Africa’s third-largest metropolis, following Lagos and Cairo, and is among the continent’s fastest-growing urban regions [15]. Kinshasa is segmented into 35 Health Zones, each of which is further divided into Health Areas. Insufficient access to water and sanitation, together with inadequate hygiene practices, malnutrition, and food insecurity, are identified as some of the primary risk factors contributing to mortality and disability in the country. The DRC reported its initial confirmed case of COVID-19 on 10 March 2020. As of 19 December 2023, there were a total of 99,333 cases, and 1468 deaths [17].

This study is grounded in a comprehensive conceptual framework of food security, which includes four key dimensions [18]: food availability, encompassing factors such as crop production, livestock holdings, and access to local markets; food access, shaped by household income, employment in informal sectors, and access to credit or financial resources; vulnerability to food shortages, impacted by exposure to economic shocks, livestock holding stability, and income levels; and utilization, which is influenced by demographic factors including age, sex, education, and household size. By examining these interconnected dimensions, the study aims to capture the complex nature of food security, particularly as it relates to the disparities between slum and non-slum areas. This framework not only supports an understanding of the individual factors contributing to food security, but also provides a structured lens for analyzing how these elements collectively influence the resilience and vulnerability of urban populations facing food insecurity.

## 2. Methods

### 2.1. Study Design

We conducted a community-based cross-sectional study spanning 27 July to 3 August 2022. The survey had a two-stage cluster sampling design. Census enumeration areas (EAs) were randomly selected in the first stage using the National Statistical Institute sampling frame. Data were collected in 62 EAs in Kinshasa. Each EA was divided into “segments” to streamline field workers’ efforts. Each segment was intended to consist of approximately 500 households to be surveyed by a specific team. The number of segments in each EA was calculated by dividing the total number of households in an EA by the average segment size of 500 households. Within a selected EA or segment, a listing of all households was obtained and used to randomly select 35 households (second stage). In the selected households, the head of the household or their designated representative was interviewed on the day of the survey. Based on standard parameters, a sample size of 2160 households with a cluster size of 35 households would produce sufficiently reliable estimates, and a five-percent reserve sample was also considered. In total, 2170 households were sampled across 62 EAs, with 35 households chosen from the EAs or segments. Finally, this study included 2020 households (response rate of 93%).

### 2.2. Data Collection

Data collection was conducted by 150 master’s students from the Kinshasa School of Public Health (KSPH), using tablets with the SurveyCTO program. They were taught research tools, ethics, and linguistic issues. The master’s students, who acted as interviewers, also received instruction on how to administer questionnaires during two training sessions conducted by the research team. Interviews were conducted in Lingala, the local language in Kinshasa, or French. We employed reverse translation, with the assistance of bilingual academics, to assure linguistic and conceptual equivalency while translating from French to Lingala. Information was gathered and examined anonymously; the survey questionnaire did not include any personal identifying information about the participants. The primary respondent at the household level was either the head of the household or their designated representative on the day of the survey. If a selected household was inaccessible or lacked a capable person to participate in the interview, interviewers were directed to make three separate visits at various times before considering the residence absent or vacant.

KSPH teaching staff that act as supervisors played a vital role in overseeing the interviewers and maintaining data quality in the field. The collected data were routinely checked by supervisors before being transmitted to the server. This process occurred every evening during the data collection phase. Supervisors conducted quality control visits to verify the correctness and completeness of data and to verify that the master’s students interviewed the appropriate respondents effectively. Quality control visits were conducted on 5% of the households in each EA. The quality check was conducted using a brief questionnaire that solely contained questions from the Household Food Insecurity Access Scale (HFIAS).

The master’s students were trained to verify the completeness and quality of their work, and the supervisors reviewed all data forms before their submissions. Forms containing omissions and clear errors were sent back to the master’s students through their supervisors for rectification or further review. The forms were checked for errors or inconsistencies at the time of data entry. Additionally, the central team conducted daily data visualization using the SurveyCTO server to provide timely feedback throughout the data collection period.

### 2.3. Measurements

#### 2.3.1. Sociodemographic Variables

The sociodemographic variables included age, sex, ethnicity, size of household, annual household income, relationship status, employment status at the time of the survey, and whether the housing was located in an informal settlement (slum) [19]. Overcrowded households were defined as those with four or more persons living in one room. Conversely, a household was deemed to have sufficient living space when three or fewer people were living in one room. The household wealth index was constructed based on principal component analysis [20] to create an index from a set of household assets (radio, tape recorder, television set, bicycle, hand torch, and horse or donkey cart), housing conditions (roof material, number of rooms, wall type, windows, availability and type of latrine), and ownership of domestic animals. The study participants were ranked according to the wealth index score, divided into quintiles, from the lowest (first quintile) to the highest (fifth quintile).

#### 2.3.2. Food Security

In alignment with the conceptual framework of food security, which emphasizes availability, access, vulnerability, and utilization, household food security status in this study was measured using the Household Food Insecurity Access Scale (HFIAS) [21]. The HFIAS, a validated instrument that differentiates food-insecure households from food-secure ones, has been utilized in prior research in the DRC [9]. The individual components of the HFIAS yield information on food insecurity (access) at the household level. Household food insecurity was measured based on nine questions along with their frequency of occurrence. The respondents were asked whether they ever: (1) worry about food, (2) are unable to eat their preferred foods, (3) eat just a few kinds of foods, (4) eat foods they would rather not eat, (5) eat smaller meals than his/her aspiration, (6) eat fewer meals in a day (less than 3 meals), (7) have no food of any kind in the household, (8) go to sleep hungry, and (9) go a full 24 h without eating. Four indicators were computed to evaluate alterations in household food insecurity (access). These variables offer an overview of household food-insecure access, including conditions and prevalence. All data were quantified such that if a household indicated “not experienced” for a specific condition, it was assigned a value of 0, and the frequency of that condition was also recorded as 0. In instances where a family encountered a certain situation, it was assigned a value of 1, while the frequency was categorized as 1 for “rarely”, 2 for “sometimes”, and 3 for “often”. Our study used the version that was previously translated into Lingala and the three other local languages to ensure accessibility and precision in measurement across diverse groups. This tool captures key domains reflecting the conceptual framework’s focus on both access and vulnerability, as it includes universal indicators of food insecurity: (1) anxiety and uncertainty about household food supply, (2) insufficient food quality (including variety and preferences for the type of food consumed), and (3) insufficient food intake and its physical consequences [22]. Household food security outcomes were categorized into: (1) food secure, (2) mildly food insecure, (3) moderately food insecure, and (4) severely food insecure, which we further dichotomized into food secure and food insecure (grouping mildly food insecure, moderately food insecure, and severely food insecure). The dependent variable (household food security status) was a dichotomous variable that was assigned a value of 1 if the household was food secure and 0 otherwise. The Cronbach’s alpha was 0.94, demonstrating excellent internal consistency.

While household food insecurity estimates provide an overall picture of a household’s access to food within the framework’s dimensions, they may not accurately reflect individual experiences within a household, which are affected by factors, such as the intra-household distribution of resources and varying dietary needs. To gain a more nuanced understanding that further aligns with the framework, combining household-level data with individual-level information or conducting separate assessments for individuals within a household may be more informative.

### 2.4. Data Analysis

All statistical analyses were conducted using STATA 17 (StataCorp, College Station, TX, USA). Initially, we determined an overview of participants’ sociodemographic characteristics, both in their entirety and as categorized by food security status. This involved employing cross-tabulations and chi-square tests to identify significant differences between non-slum and slum households. Significant differences in food security status were evaluated using chi-square tests. To assess household susceptibility to food insecurity as the primary outcome variable, a multivariable logistic regression model was developed. Factors associated with food insecurity in bivariate analysis were entered into a logistic regression model to obtain adjusted odds ratios (AORs) and their 95% confidence intervals (95% CIs). To assess how the association between slum neighborhoods and food insecurity might differ by living space status, an interaction term between living space status and slum residence was included in the multivariable model, and the log-likelihood ratio test was used to assess its significance.

The Breslow–Day test for assessing the interaction effect was used. If it was found to be significant at *p* < 0.05; separate multivariate regression analyses were performed by type of neighborhood. The interactions between living space status and slum residence, between living space status and wealth index, and between wealth index and slum residence did not suggest heterogeneity between slum and non-slum residents. All of the statistical analyses were conducted using Stata Version 17.0. SVY procedures in Stata were used to account for the sampling design and selection weights. ORs and 95% CIs were estimated from the regression parameters. Variance inflation factors were calculated to test for multicollinearity, with the highest found to be 2.65. The significance level was set at *p* < 0.05. We used the STROBE cross-sectional reporting guidelines.

### 2.5. Ethical Considerations

This study was conducted in accordance with the principles outlined in the Helsinki Declaration. The master’s students were trained to obtain informed consent. Respondents who could not express themselves in French were offered consent forms in their preferred language. Ethical clearance was obtained from the ethical review board of the KSPH (no. ESP/CE/71B/2022). Each participant had the informed consent form read aloud to them and provided verbal consent. To standardize the informed consent process for illiterate participants, we opted for verbal consent witnessed by a third party. The third party ensured that the consent was read to a participant, who then freely agreed to participate in the study. The participant was provided with a signed copy of the consent form to retain. The consent acquisition process was sanctioned by the Ethics Committee. This study did not involve any individuals under the age of 18. Information was gathered and examined anonymously. The survey form did not include any personal identifying information about participants, and the participants were notified that their involvement was optional. They had the liberty to either accept, decline participation, or withdraw at any time without facing any consequences.

### 2.6. Participant and Public Involvement

During the pilot phase of this study, the research team actively listened to and addressed participants’ perspectives on the significance of evaluating food security. Participants were also given the opportunity to provide feedback on the research methodologies and processes. This study will be distributed to important individuals and organizations (such as policymakers, implementing partners, and a representative sample of the community) in order to engage them in the utilization of the evidence generated.

## 3. Results

### 3.1. Participants’ Characteristics

Overall, most individuals interviewed identified as the head of the household and were at least 40 years old (72%). Almost three-quarters of the participants were men (73.6%) and married (72.5%), and 60% had a schooling level between secondary school and university. Table 1 illustrates that the mean household size was six persons or fewer (75%); approximately one-third of the households were overcrowded (29%), and the socioeconomic status (SES) distribution among households was not equal. For these characteristics, the differences between people living in non-slums and people living in slums were statistically significant.

### 3.2. Food Security Conditions of Households

Table 2 presents the proportion of households that experienced food insecurity in a particular condition, as well as the frequency of experiencing that condition. The results reveal significant differences in the proportion of households that experienced the condition at any time during the recall period between non-slum and slum households across. The results reveal that 66% of the households were worried about food insecurity (non-slum: 59.0, slum: 69.6; *p* < 0.001) and that 9% of them experienced this condition more than 10 times during the last four weeks as of the survey period. Around 64% of the households were unable to eat their preferred foods, and 10% frequently experienced this condition. Limited varieties of foods were consumed by 63% of the households, and this condition was frequently experienced by 10% of the households. Nearly 65% of the households consumed foods that they would prefer not to eat, and almost 62% consumed smaller meals during the last four weeks as of the study period. Fewer than three meals per day were eaten by 60% of the households, and 9% of them experienced this condition regularly. About 6% of the households frequently had no food to eat at home, and 5% went to sleep hungry or regularly went a full day without eating. 

### 3.3. Household Food Insecurity Access Prevalence

The prevalence of household food insecurity access was categorized into four groups. In total, a mere 23.5% (with a 95% confidence interval ranging from 21.6 to 25.4) of households were found to be food secure. The prevalence rates of severe food insecurity, moderate food insecurity, and mild food insecurity were 55.9%, 15.2%, and 5.4%, respectively. The prevalence of food insecurity was 76.5% (95%CI: 74.6–78.3). Figure 1 is presenting the distribution of food insecurity based on household characteristics.

### 3.4. Factors Associated with Food Insecurity

Figure 2 illustrates the factors associated with food insecurity. Specifically, individuals aged 50 years or older exhibited a significantly greater likelihood of experiencing food insecurity (AOR: 2.02; 95% confidence interval (CI): 1.23–3.31) as compared to their counterparts. Conversely, having a sufficient living space was associated with a reduced likelihood of food insecurity (AOR: 0.56; 95% CI: 0.44–0.77). Additionally, households characterized by lower SES (AOR lowest: 5.36; 95% CI: 3.29–8.74; AOR low: 3.30; 95% CI: 2.19–4.98; AOR moderate: 1.97; 95% CI: 1.37–2.84; and AOR higher: 1.48; 95% CI: 1.06–2.06) as compared to their counterparts. Living in a slum area (AOR: 1.38; 95% CI: 1.06–1.79) was associated with food insecurity.

## 4. Discussion

This study assessed household food insecurity during the COVID-19 pandemic at the household level in Kinshasa, focusing on the heterogeneity between slum and non-slum areas and identifying associated factors. Based on the study’s framework, availability is tied to factors such as crop production and market access; access involves household income and employment type; vulnerability reflects exposure to economic shocks; and utilization is influenced by household composition, sanitation, and education levels. Our findings reinforce this model, showing that economic instability during the pandemic affected all dimensions of food security and disproportionately impacted slum households. 

The study observed high levels of food insecurity, with over half of the households affected, particularly in slum areas. This disparity was strongly associated with social and environmental variables, such as age, SES, and household living conditions, all of which interplay with broader food security factors. Most household heads were at least 40 years old (72%), and SES was evenly distributed. However, our results reveal that those aged 50 years or older, having insufficient living space, having a certain SES (poorest, poorer, middle, or wealthier), and residing in a slum area faced a heightened risk of food insecurity. 

Such findings reinforce the food security framework’s emphasis on access and vulnerability to economic shocks (such as COVID-19 lockdowns) as key dimensions, illustrating how specific socioeconomic profiles shape food insecurity levels. These findings align with prior research demonstrating that economic shocks can lead to decreased food accessibility in vulnerable households; thus, the progression of the pandemic made it possible for food security to deteriorate further [2,4,6,8,9,10,23,24]. Lockdown measures associated with the pandemic could have disrupted food supply chains and dietary habits, potentially contributing to various forms of malnutrition, including an increased risk of obesity owing to the consumption of highly processed foods and reduced physical activity [2]. Therefore, it is essential to closely monitor the indirect health effects of the COVID-19 pandemic. 

The COVID-19 pandemic significantly affected food insecurity in Africa, intensifying existing difficulties and introducing new ones. Several factors illustrate COVID-19′s impact on food security [25]: border closures and limitations impeded the transportation of products, notably food, across borders, resulting in shortages and price escalations. Lockdowns and movement restrictions impeded the transfer of food from production regions to markets, affecting supply networks. Numerous agricultural laborers were incapacitated from working owing to lockdowns, illness, or apprehension of infection, adversely affecting productivity and harvesting. In Africa, the pandemic heavily impacted informal jobs and small enterprises, causing job losses and reduced incomes, limiting food affordability [26]. Remittances from expatriate workers, an essential economic source for numerous African families, decreased as global economies contracted. The economic recession induced by the epidemic exacerbated poverty, hence restricting individuals’ access to food. The pandemic resulted in heightened demand for specific food items, especially basics, leading to an increase in prices.

As per the utilization dimension, food insecurity was more prevalent in informal settlement households, in which more than three-quarters of the households reported having experienced this issue. Households lacking consistent access to cooking fuel, medical care, electricity, or water and sanitation, in addition to cash income, demonstrated significantly higher odds of being categorized as food insecure. Limited access to food resources exacerbates the adverse effects of the pandemic and obstructs infection control measures. Our study’s results emphasize the critical importance of addressing inequities in accessing physical and social infrastructure for food security beyond household income, which aligns with Frayne and McCordic’s findings that urban environments lacking in essential services in South Africa demonstrated higher food insecurity risks [27].

### 4.1. Policy and Practice Implications

The findings underscore the necessity of policies informed by a comprehensive security framework that not only address immediate food needs, but also strengthen economic stability and infrastructure to build resilience in slum communities. 

First, focused social protection actions are needed against economic shock, accounting for household profiles [2,3,8]. Vulnerable households, particularly those with informal jobs and young children, could benefit from cash transfer strategies to protect their food security. 

Second, to address vulnerability, it is imperative to establish robust mechanisms to monitor the food insecurity status of households during shock events, paying particular attention to vulnerable groups, such as older individuals and households relying on informal employment [3,8]. A study conducted in Mexico during the COVID-19 pandemic recommended monitoring food insecurity in the general population, including critical vulnerable groups such as those with low and middle SES [2,28,29]. However, limitations of the SES scale should be considered, as it may not fully capture changes in economic circumstances, instead reflecting only pre-pandemic SES levels. 

Third, the expansion of food assistance programs, including cash transfers and food supply initiatives, should target overcrowded households and the informal job sector. Providing and evaluating not only food assistance but also cash transfer initiatives, at least for the most vulnerable households, is important. A cash transfer strategy for households with informal jobs and children younger than five years old may help protect their food security status. 

Fourth, to enhance crisis resilience in low-income countries, investments should prioritize strengthening infrastructure, advancing female empowerment, boosting economic performance, developing human capital, and establishing an agile emergency workforce. Such initiatives are critical to mitigating the impact of crises, including natural disasters and public health emergencies, and fostering long-term stability and recovery, as highlighted by Khan et al. [30]. 

Fifth, policymakers should consider interventions that enhance food availability, such as supporting small-scale farmers by providing access to modern agricultural technologies, education, financing, and market opportunities, thereby improving food availability, enhancing livelihoods, and contributing to sustainable agricultural development. Policies that encourage sustainable agriculture, protect land tenure rights, and foster investment are essential for long-term productivity and stability. Additionally, transferring applicable agricultural technologies from wealthier nations to an African context can bolster agricultural efficiency.

Sixth, addressing interconnected syndemic-like crises warrants a multifaceted approach guided by data-driven strategies grounded in a flexible and intricate systems framework [5]. Importantly, even as the COVID-19 pandemic recedes, its far-reaching economic, health, and societal consequences are expected to persist over an extended period [11].

### 4.2. Study Strengths and Limitations

This study had several strengths, including the comprehensive assessment of food insecurity on a large scale during the COVID-19 pandemic, which allows for generalizability across all of Kinshasa and other similar urban centers in the DRC and other sub-Saharan countries characterized by the presence of slums and non-slum areas. However, certain limitations must be acknowledged. First, this study did not assess the broader multisectoral impacts of COVID-19 beyond its influence on food security. Additionally, the pre-pandemic food security conditions of households were not evaluated, thereby preventing direct comparisons of conditions before and during the pandemic. We did not provide a qualitative description of the food system of households in Kinshasa, including its primary characteristics, food typologies, and dishes in both slum and non-slum areas. In this study, evaluation of confounding was particularly relevant in the identification of factors associated with food insecurity. Most of the confounders were in fact controlled during the data analysis stage, using multivariate analyses. Even if we used a multivariate technique, there could remain residual confounding in this study. There was the probability that the additional confounding factors were not considered, because data on these factors were not collected. Self-reported data and recall bias must be considered when interpreting the study results. This study did not assess the link between poor sanitation and food utilization or how lack of access to clean water in slums contributes to poor nutritional outcomes.

Despite employing random sampling, we lack the means to confirm the representativeness of our sample in relation to the population of Kinshasa. The DRC last conducted a census in 1984, and there is no clear indication of whether our sample accurately represents the population of Kinshasa. Gender considerations, albeit rarely openly addressed in the DRC, could impact food security in Kinshasa. This study is also limited by a lack of information collected about gender. Future research should analyze how food insecurity and the factors contributing to it develop, particularly in response to the economic consequences of the COVID-19 pandemic. The potential confounding influence of certain variables on food security, which were not collected in this study, cannot be ruled out. Social and structural determinants potentially contributing to food security, such as female empowerment and access to water, electricity, and medical care, should be further researched. Age categorization may also lead to residual confusion.

## 5. Conclusions

Aligned with the theoretical framework, our study concludes that the COVID-19 pandemic significantly impacted household food security in Kinshasa, with potentially severe implications for vulnerable households, older individuals, and informal workers. Addressing these challenges requires the government to develop targeted strategies aimed at mitigating household vulnerability during crises. Food availability, access, and vulnerability—all crucial dimensions of food security—were disrupted, emphasizing the need for resilience-building policies in affected communities.

Effective interventions should support small-scale farmers with modern agricultural technologies, education, and financial resources, enhancing local food availability. Policies that secure land tenure and promote sustainable agriculture can improve productivity and strengthen long-term food security. Cross-national collaboration to transfer relevant agricultural technologies can also reinforce food systems in vulnerable regions.

For immediate relief, expanding food assistance programs such as cash transfers and targeted food supplies is critical, with a focus on overcrowded and informal job sector households. Continuous monitoring of food insecurity among high-risk groups will enable data-driven, timely responses. By integrating these multi-faceted approaches, governments can create a resilient food system and support urban populations’ health and well-being, particularly in cities such as Kinshasa.

## Figures and Tables

**Figure 1 foods-13-03657-f001:**
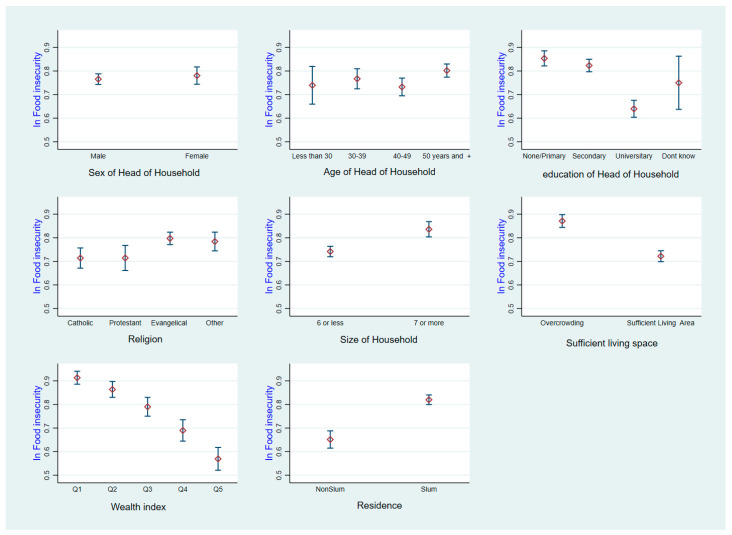
Food insecurity prevalence by household characteristics.

**Figure 2 foods-13-03657-f002:**
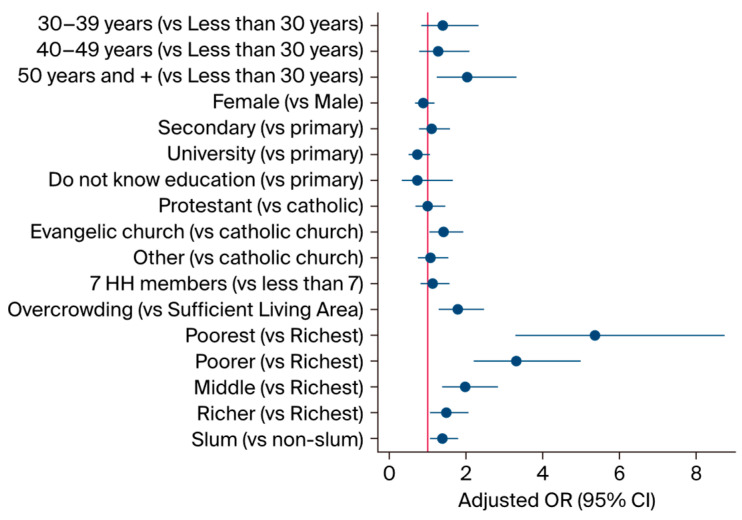
Forecast plot presenting the associated factors of food insecurity with their adjusted ORs.

**Table 1 foods-13-03657-t001:** Sociodemographic characteristics of the heads of the households.

	Non-Slum		Slum		All		*p*
	n	%	n	%	n	%	
Age							0.555
Younger than 30 years	39	7.0	80	6.3	119	6.5	
30–39	107	19.2	275	21.7	382	20.9	
40–49	161	28.9	377	29.7	538	29.5	
50 years or older	250	44.9	537	42.3	787	43.1	
Total	557	100.0	1269	100.0	1826	100.0	
Sex							0.001
Male	387	68.5	982	75.8	1369	73.6	
Female	178	31.5	314	24.2	492	26.4	
Total	565	100.0	1296	100.0	1861	100.0	
Marital status							0.037
Married	457	69.9	1007	73.7	1464	72.5	
Divorced/separated	58	8.9	121	8.9	179	8.9	
Widowed	80	12.2	161	11.8	241	11.9	
Single	59	9.0	77	5.6	136	6.7	
Total	654	100.0	1366	100.0	2020	100.0	
Schooling							<0.001
Nothing/primary	78	11.9	393	28.8	471	23.3	
Secondary	224	34.3	574	42.0	798	39.5	
University	338	51.7	353	25.8	691	34.2	
No answer	14	2.1	46	3.4	60	3.0	
Total	654	100.0	1366	100.0	2020	100.0	
Religion							<0.001
Catholic	178	27.2	252	18.4	430	21.3	
Protestant	112	17.1	168	12.3	280	13.9	
Evangelic	261	39.9	632	46.3	893	44.2	
Other	103	15.7	314	23.0	417	20.6	
Total	654	100.0	1366	100.0	2020	100.0	
Size of household							<0.001
Six persons or fewer	548	83.8	966	70.7	1514	75.0	
Seven persons or more	106	16.2	400	29.3	506	25.0	
Total	654	100.0	1366	100.0	2020	100.0	
Sufficient living space (Not overcrowded) *							<0.001
Sufficient living space	515	78.8	916	67.1	1431	70.8	
Overcrowding	139	21.2	450	32.9	589	29.2	
Quintile of SES							<0.001
Lowest	44	6.7	360	26.4	404	20.0	
Lower	74	11.3	330	24.2	404	20.0	
Middle	124	19.0	281	20.6	405	20.0	
Higher	159	24.3	244	17.9	403	20.0	
Highest	253	38.7	151	11.1	404	20.0	
Total	654	100.0	1366	100.0	2020	100.0	

Note: * Three or fewer persons per living room: sufficient living space; more than three persons per living room: overcrowding. SES: socioeconomic status.

**Table 2 foods-13-03657-t002:** Food insecurity access-related condition of the studied households by residence.

HFIAS Questions	Households Experienced the Condition at Any Time During the Recall Period (%)				Households Experienced the Condition (Often) at a Given Frequency (%)			
	Non-Slum (n = 654)	Slum (n = 1366)	All (n = 2020)	*p*	Non-Slum (n = 654)	Slum (n = 1366)	All (n = 2020)	*p*
Worry about food	59.0	69.6	66.2	<0.001	7.0	9.4	8.7	0.072
Unable to eat preferred foods	52.0	69.3	63.7	<0.001	8.1	10.4	9.7	0.103
Eat a limited variety of foods	50.6	69.5	63.4	<0.001	7.5	11.0	9.9	0.014
Eat foods that they really did not want to eat	53.2	70.6	65.0	<0.001	8.3	11.5	10.4	0.026
Eat a smaller meal	49.8	68.1	62.2	<0.001	6.3	10.5	9.1	0.002
Eat fewer meals in a day	47.4	65.6	59.7	<0.001	7.8	9.2	8.8	0.289
No food to eat of any kind in the household	35.8	55.8	49.3	<0.001	3.7	6.9	5.8	0.004
Went to sleep at night hungry	31.0	50.4	44.1	<0.001	3.7	5.9	5.2	0.032
Went a whole day and night without eating anything	29.8	47.2	41.6	<0.001	3.4	4.2	4.0	0.342

## Data Availability

All the relevant data for this study are available from the KSPH. The materials will be made available by the leading author upon request. The dataset can be found also at osf: https://osf.io/h4jt2/?view_only=ad40b39d0f584bf480fb04571ed86b70 (accessed on 1 August 2024).

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
