# Peer review of "Household Food Insecurity During the COVID-19 Pandemic Between Slum and Non-Slum Areas in Kinshasa, DR Congo: A Cross-Sectional Study"

_foods, 2024, doi:10.3390/foods13223657_

Round 1
Reviewer 1 Report
Comments and Suggestions for Authors
This is a very well-written and well organized paper based upon an in-depth survey investigating food insecurity in Kinshasa.
The adopted methods are sound and the findings very relevant.
I would just encourage the authors to improve the paper according to the following suggestions:
1. Introduction. More has to be written about food insecurity addressed by previous field studues in diverse urban African contexts and why these studies are important
2. Results. Authors should even only qualitatively describe the food system of Kinshasa's households, its main characteristics, food typologies, dishes both in slums and non slums.
3. Discussion. Authors should tell more about the diachronic situation ore and post Covid and interpretat the data, i.e. describe the hypothetical mechanisms through which insecurity increased.
Moreover, they should discuss more possible concrete solutions that can be very concretely afopted by local and national policy makers.
Comments on the Quality of English LanguageGood
Author Response
Comments and Suggestions for Authors
This is a very well-written and well-organized paper based upon an in-depth survey investigating food insecurity in Kinshasa.
The methods adopted are sound and the findings very relevant.
I would just encourage the authors to improve the paper according to the following suggestions:
- Introduction. More has to be written about food insecurity addressed by previous field studies in diverse urban African contexts and why these studies are important
Authors response: We have provided short justification which can be seen as follows:
“Inhabitants of slums sometimes endure severe poverty, restricted access to resources, and unstable living conditions. These conditions render individuals particularly vulnerable to food insecurity. Non-slum communities, despite superior infrastructure and resource accessibility, may nonetheless encounter food insecurity due to reasons such as income disparity, unemployment, and escalating food prices. Examining slum regions might uncover distinct difficulties such as inadequate sanitation, scarcity of clean water, and restricted food storage alternatives, which can intensify food insecurity. Comprehending these particular difficulties might facilitate the formulation of focused interventions and policies to successfully address the needs of slum inhabitants. Moreover, examining food insecurity in both slum and non-slum areas can facilitate the identification of at-risk groups and evaluate the overall resilience of urban food systems. Examining the disparity between slum and non-slum regions, we aim also to attain a more profound comprehension of the intricate elements that contribute to food insecurity in urban setting and formulate effective solutions to tackle this urgent concern.”
- Results. Authors should even only qualitatively describe the food system of Kinshasa's households, its main characteristics, food typologies, dishes both in slums and non-slums.
Authors response: We did not collect qualitative data, which is undoubtedly a shortcoming of this study, thanks to this insightful criticism. which can be seen as follows: “We did not provide a qualitative description of the food system of households in Kinshasa, including its primary characteristics, food typologies, and dishes in both slum and non-slum areas”
- Discussion. Authors should tell more about the diachronic situation ore and post Covid and interpretat the data, i.e. describe the hypothetical mechanisms through which insecurity increased.
Authors response: We have addressed this in the revised version, which can be seen as follows: “The COVID-19 pandemic significantly affected food insecurity in Africa, intensifying existing difficulties and introducing new ones. Here are eight factors that elucidate the impact of COVID-19: Border closures and limitations impeded the transportation of products, notably food, across borders, resulting in shortages and price escalations. Lockdowns and movement restrictions impeded the transfer of food from production regions to markets, affecting supply networks. Numerous agricultural laborers were incapacitated from working owing to lockdowns, illness, or apprehension of infection, adversely affecting productivity and harvesting. Numerous individuals in Africa depend on informal jobs or small enterprises, which were significantly impacted by the pandemic. This resulted in job losses and diminished income, complicating individuals' ability to afford food. Remittances from expatriate workers, an essential economic source for numerous African families, decreased as global economies contracted. The economic recession induced by the epidemic exacerbated poverty, hence restricting individuals' access to food. The pandemic resulted in heightened demand for specific food items, especially basics, leading to an increase in prices”
Moreover, they should discuss more possible concrete solutions that can be very concretely afopted by local and national policy makers.
Authors response: We have addressed this in the revised version, which can be seen as follows: “Through the implementation of specific interventions, households can markedly diminish food insecurity, enhance livelihoods, and attain sustainable agricultural development. Assist small-scale farmers by providing access to contemporary agricultural technologies, education, financing, and market opportunities. Formulate policies that endorse sustainable agriculture, enhance food security, and stimulate investment. Ensure secure land tenure rights for farmers to promote investment and enhance productivity. Facilitate the transmission of agricultural technologies from wealthy nations to African countries.”

Reviewer 2 Report
Comments and Suggestions for Authors
I have read with interest the article entitled:
Adaptation and validation of the Child Eating Disorder Examination-Questionnaire (ChEDE-Q) for use in English among adolescents in urban India ». The title which describes the article and the abstract as well as the introduction and the objectives are clearly laid out . Τhe methodology is well described
The results are clearly led out and in a logical sequence. The statistics used are appropriate for this type of data. The conclusions are well discussed, the righting is excellent and the figures inform the reader and are an important part of the study.
I have only a minor remark: Page 3 line 24, please give details about sample size requirements .
Author Response
Adaptation and validation of the Child Eating Disorder Examination-Questionnaire (ChEDE-Q) for use in English among adolescents in urban India ». The title which describes the article and the abstract as well as the introduction and the objectives are clearly laid out . Τhe methodology is well described
Authors response: This is not the title of our manuscript.
The results are clearly led out and in a logical sequence. The statistics used are appropriate for this type of data. The conclusions are well discussed, the righting is excellent and the figures inform the reader and are an important part of the study.
I have only a minor remark: Page 3 line 24, please give details about sample size requirements .
Authors response: The details about sample size requirements are provided in the manuscript

Reviewer 3 Report
Comments and Suggestions for Authors
The paper addresses a critical issue, but it would benefit from a stronger theoretical foundation, clearer methodological explanations, improved logical flow between results and recommendations, and a more comprehensive discussion of limitations. Additionally, the study should explore the role of gender and other social determinants in shaping food insecurity outcomes. By addressing these issues, the study will not only present a clearer analysis but also offer more actionable insights for policymakers and practitioners.
1. Lack of Theoretical Basis
The paper addresses food insecurity in urban slums and non-slums but lacks a strong theoretical foundation. The literature review does not adequately cover relevant frameworks or theories, such as food security, urban poverty, and public health. Furthermore, the relationships between food security, socioeconomic factors, and health outcomes are not well explained, resulting in an underdeveloped theoretical base for the study.
Suggestions: The theoretical foundation needs to be expanded, particularly by incorporating well-established food security frameworks, such as the FAO’s four dimensions of food security: availability, access, utilization, and stability. These dimensions should be discussed in the context of how urbanization, income inequality, and informal economies affect food security. Additionally, the paper could include a discussion on how the post-pandemic economy may have exacerbated food insecurity and how social structures and economic recovery models relate to the issue. This would provide a stronger theoretical framework to guide the study’s analysis and interpretation of the results.
2. Insufficient Empirical Methods
The paper employs cross-sectional data and logistic regression models, but the description of the empirical methods is insufficient. The sampling strategy, variable operationalization, and potential selection biases are not well elaborated. The definition of key variables, such as "socioeconomic status" and "overcrowding," lacks detail, and this lack of clarity can lead to misinterpretation of results. The measurement tools and methods of categorization are not clearly defined, which reduces the transparency of the methodology.
Suggestions: The methodology section should provide a detailed explanation of the sampling process, including how the surveyed areas were selected and how representativeness was ensured. A clearer explanation of how key variables were operationalized is essential. For instance, the criteria for classifying "socioeconomic quintiles" and determining "overcrowding" need to be defined clearly, possibly referencing widely accepted standards. This will enhance the robustness and replicability of the study. Furthermore, the analysis could benefit from a discussion of the potential biases that could arise from the sampling or data collection methods, such as selection bias or response bias, and how the study design addresses or mitigates these concerns.
3. Logical Issues in Argumentation
The logical flow of the paper is inconsistent, particularly between the results and recommendations. The authors identify several factors associated with food insecurity, such as age, overcrowding, and socioeconomic status, but the link between these factors and the policy recommendations is weak. For instance, the paper suggests interventions like cash transfers and food assistance programs, but these are not clearly connected to the empirical findings.
Suggestions: The paper needs a more cohesive structure, with clear logical connections between the empirical results and the recommendations. For example, if overcrowding is a key factor linked to food insecurity, the recommendations should specifically address how improving housing conditions could help alleviate food insecurity. Similarly, interventions like cash transfers should be explicitly tied to the analysis of income or socioeconomic data. Each recommendation should flow directly from the study’s findings, ensuring the argument is clear and well-supported throughout the paper.
4. Weak Connection Between Results and Recommendations
While the paper identifies key predictors of food insecurity, such as poverty, overcrowding, and slum residence, the proposed policy recommendations seem generic and not directly derived from the results. For instance, the suggestion of cash transfers lacks a clear explanation of how it would address the specific causes of food insecurity identified in the study, such as poor housing conditions or informal employment. The disconnect between findings and suggestions reduces the practical relevance of the recommendations.
Suggestions: Each policy recommendation should be directly tied to the study’s specific findings. For example, if socioeconomic status is a significant predictor of food insecurity, the recommendations should focus on targeted poverty alleviation strategies, such as income support or employment programs that directly address income disparities. Similarly, if overcrowding is a key issue, policies aimed at improving urban infrastructure and housing conditions should be considered. The recommendations should provide actionable strategies that directly address the factors identified in the empirical analysis.
5. Incomplete Discussion of Study Limitations
Although the paper acknowledges some limitations, such as the lack of pre-pandemic data, the discussion is not comprehensive. It overlooks several important factors that could have influenced the results, such as differences in wealth distribution, education levels, and access to services between slum and non-slum areas. Additionally, the study relies heavily on self-reported data, which is prone to recall and reporting bias, yet this issue is not adequately addressed.
Suggestions: A more thorough discussion of the study’s limitations is needed. The authors should acknowledge the limitations of using self-reported data, such as recall bias, and discuss how this could have impacted the results. Furthermore, the limitations of the cross-sectional study design should be addressed, as this approach only allows for the identification of associations and not causality. The authors could suggest using longitudinal data or mixed-method approaches in future research to strengthen causal inferences and address the limitations of the current study design.
6. Lack of Gender Analysis
Despite focusing on household food insecurity, the paper does not explore gender differences, which are often significant in studies of food insecurity. Women and children, especially in urban poor settings, are more likely to experience food insecurity, yet the paper does not investigate whether female-headed households face higher risks or how gender roles within households may impact food distribution.
Suggestions: Incorporating gender as a key variable could provide valuable insights into the dynamics of food insecurity in slums and non-slums. The authors should analyze whether female-headed households are more vulnerable to food insecurity and examine the role of gender in household food distribution. This could also lead to more targeted recommendations, such as empowering women through livelihood programs or increasing access to social services for female-headed households. Including a gender analysis would strengthen the study and contribute to a more nuanced understanding of the factors influencing food insecurity.
7. Lack of Specificity in Policy Recommendations
The paper’s policy recommendations are broad and lack concrete implementation strategies. For example, while cash transfers and food assistance are proposed as solutions, the recommendations do not include details about how these programs would be rolled out, who would be targeted, or how effectiveness would be monitored. Without these specifics, the recommendations appear less practical and actionable.
Suggestions: The recommendations should be more specific and actionable. For instance, the paper could outline how cash transfer programs could be implemented in slum areas, with clear criteria for identifying beneficiaries and mechanisms for delivering aid. The authors could also discuss potential challenges, such as administrative capacity or targeting efficiency, and suggest ways to address these. Drawing on examples from other countries or regions where similar interventions have been successfully implemented would provide empirical support for the recommendations and make them more credible.
8. Limited Cross-Sectoral Discussion
Although the paper discusses food insecurity, it does not sufficiently address the broader contextual factors that contribute to this issue, such as infrastructure, access to clean water, sanitation, and public health services. These factors are crucial in understanding the full scope of food insecurity, particularly in urban slums where overcrowding and poor living conditions exacerbate the problem.
Suggestions: The authors should expand the scope of the discussion to include how cross-sectoral factors such as urban infrastructure, public health, and sanitation affect food security. For instance, they could explore the link between poor sanitation and food utilization or how lack of access to clean water in slums contributes to poor nutritional outcomes. By addressing these broader factors, the paper would provide a more comprehensive understanding of the challenges of food insecurity and offer more holistic policy solutions that tackle the issue from multiple angles.
Comments on the Quality of English LanguageSeveral grammar issues have been identified that impact the overall clarity and readability of the paper. Below is a comprehensive explanation of these issues and recommendations for improvement.
One of the most noticeable grammar problems in the paper is the inconsistency in verb tense. The paper frequently shifts between past, present, and future tenses, often within the same paragraph or sentence. This inconsistency makes it difficult for the reader to follow the timeline of the research or understand whether the statements refer to past studies, present circumstances, or future implications. For instance, when discussing research that has already been conducted, the past tense should be consistently used, while general statements of fact can be written in the present tense. Maintaining tense consistency throughout the paper will create a more coherent and professional narrative.
Another issue is subject-verb agreement. In many cases, plural subjects are mistakenly paired with singular verbs, and vice versa. This type of error can confuse the reader and detract from the academic tone of the paper. For example, in sentences where the subject is plural, such as "the data from the surveys," the verb must also be plural to match, as in "the data from the surveys show." Ensuring that subjects and verbs agree in number throughout the paper is essential for clarity and correctness.
Additionally, the paper contains both sentence fragments and run-on sentences, which can affect the flow of ideas and make the text harder to comprehend. Sentence fragments occur when incomplete thoughts are presented without a main clause, while run-on sentences happen when multiple independent clauses are joined without proper punctuation or conjunctions. To resolve these issues, it is important to structure sentences so that each expresses a complete idea and uses appropriate punctuation to separate or connect thoughts. For instance, joining two independent clauses with a conjunction like “and” or “but” or using a semicolon to link closely related ideas can improve the overall sentence structure.
Redundancy and wordiness also appear frequently in the text, where ideas are unnecessarily repeated or expressed in a verbose manner. This diminishes the paper's clarity and can cause the writing to feel repetitive or overly complicated. To address this, aim to express ideas more concisely and avoid restating the same concept multiple times within a single section. For example, instead of stating, "It is very clear and evident that there is a strong and significant relationship between income level and food insecurity," a more concise version would be, "There is a significant relationship between income level and food insecurity."
The misuse of articles (a, an, the) is another common issue found in the document. The incorrect or inconsistent use of articles can alter the meaning of a sentence or make it grammatically incorrect. For instance, phrases like "due to lack of proper housing" should be corrected to "due to **a** lack of proper housing." When referring to specific or previously mentioned nouns, the definite article "the" should be used, whereas indefinite articles "a" or "an" should be used for more general or singular references.
Another issue involves unclear or ambiguous pronoun references. In several instances, pronouns like “it” or “they” do not clearly refer to a specific noun, which can confuse the reader about what or who is being discussed. To improve clarity, make sure that every pronoun has a clear antecedent. When necessary, it’s better to restate the noun to avoid confusion, particularly when several potential antecedents are present in the same sentence.
There are also instances of incorrect preposition use throughout the text. Prepositions such as "on," "at," "in," and "by" must be used correctly to maintain grammatical accuracy and convey the intended meaning. For example, phrases like "aimed on improving food security" should be corrected to "aimed **at** improving food security." Ensuring proper preposition use throughout the paper will help eliminate awkward phrasing and make the arguments more precise.
The overuse of passive voice is another concern. While the passive voice is acceptable in academic writing, an excessive use of it can make the text less direct and harder to follow. In many cases, using the active voice can make sentences clearer and more engaging. For example, instead of saying “It was found that many households were food insecure,” the active voice could be used to say “The survey found that many households were food insecure.” This switch to the active voice emphasizes the subject and clarifies who or what is performing the action.
Another problem identified is the presence of comma splices, where two independent clauses are incorrectly joined by a comma without the proper use of a coordinating conjunction. Comma splices create confusion and detract from the sentence’s readability. For instance, a sentence like “The economy in Kinshasa is struggling, food insecurity continues to rise” should be corrected by adding a conjunction: “The economy in Kinshasa is struggling, and food insecurity continues to rise.” Alternatively, a period could be used to split the clauses into two distinct sentences.
Finally, some instances of incorrect word choice were identified in the text. Misusing certain words can distort the meaning of a sentence or make it sound awkward. For example, using the phrase "detrimental to" instead of "negatively impacts" may not be the most appropriate choice in certain contexts. Choosing more precise words will enhance the clarity and academic tone of the writing. It’s also important to use vocabulary that accurately conveys the intended meaning, avoiding unnecessary complexity or ambiguity.
By addressing these grammatical issues—such as verb tense consistency, subject-verb agreement, sentence structure, redundancy, article use, pronoun clarity, preposition choice, voice (active vs. passive), comma splices, and word choice—the overall quality, coherence, and professionalism of the paper can be significantly improved. Paying attention to these details will not only enhance readability but also ensure that the arguments and findings are communicated clearly and effectively.
Author Response
Comments and Suggestions for Authors
The paper addresses a critical issue, but it would benefit from a stronger theoretical foundation, clearer methodological explanations, improved logical flow between results and recommendations, and a more comprehensive discussion of limitations. Additionally, the study should explore the role of gender and other social determinants in shaping food insecurity outcomes. By addressing these issues, the study will not only present a clearer analysis but also offer more actionable insights for policymakers and practitioners.
- Lack of Theoretical Basis
The paper addresses food insecurity in urban slums and non-slums but lacks a strong theoretical foundation. The literature review does not adequately cover relevant frameworks or theories, such as food security, urban poverty, and public health. Furthermore, the relationships between food security, socioeconomic factors, and health outcomes are not well explained, resulting in an underdeveloped theoretical base for the study.
Suggestions: The theoretical foundation needs to be expanded, particularly by incorporating well-established food security frameworks, such as the FAO’s four dimensions of food security: availability, access, utilization, and stability. These dimensions should be discussed in the context of how urbanization, income inequality, and informal economies affect food security.
Authors response: We have provided short justification which can be seen as follows:
“Urbanization, wealth disparity, and informal economies are intricate elements that can profoundly influence food security, both favorably and unfavorably. Urban growth frequently intrudes upon agricultural land, diminishing food production capability. Urban populations generally exhibit elevated food consumption habits, exerting strain on food supplies. Urban regions may become too dependent on food imports, rendering them susceptible to price volatility and supply interruptions. Accelerated urbanization may result in the proliferation of informal settlements characterized by restricted access to affordable and healthy food. Income inequality can create a substantial disparity in access to healthy food, especially for low-income households. Low-income households are particularly vulnerable to price volatility, as a greater proportion of their income is allocated to food expenditures. Inhabitants of slums sometimes endure severe poverty, restricted access to resources, and unstable living conditions. These conditions render individuals particularly vulnerable to food insecurity. Non-slum communities, despite superior infrastructure and resource accessibility, may nonetheless encounter food insecurity due to reasons such as income disparity, unemployment, and escalating food prices. Examining slum regions might uncover distinct difficulties such as inadequate sanitation, scarcity of clean water, and restricted food storage alternatives, which can intensify food insecurity. Comprehending these difficulties might facilitate the formulation of focused interventions and policies to successfully address the needs of slum inhabitants. Moreover, examining food insecurity in both slum and non-slum areas can facilitate the identification of at-risk groups and evaluate the overall resilience of urban food systems. Examining the disparity between slum and non-slum regions, we aim also to attain a more profound understanding of the intricate elements that contribute to food insecurity in urban settings and formulate effective solutions to tackle this urgent concern.”
Additionally, the paper could include a discussion on how the post-pandemic economy may have exacerbated food insecurity and how social structures and economic recovery models relate to the issue. This would provide a stronger theoretical framework to guide the study’s analysis and interpretation of the results.
We have addressed this in the revised version, which can be seen as follows: “The COVID-19 pandemic significantly affected food insecurity in Africa, intensifying existing difficulties and introducing new ones. Here are eight factors that elucidate the impact of COVID-19: Border closures and limitations impeded the transportation of products, notably food, across borders, resulting in shortages and price escalations. Lockdowns and movement restrictions impeded the transfer of food from production regions to markets, affecting supply networks. Numerous agricultural laborers were incapacitated from working owing to lockdowns, illness, or apprehension of infection, adversely affecting productivity and harvesting. Numerous individuals in Africa depend on informal jobs or small enterprises, which were significantly impacted by the pandemic. This resulted in job losses and diminished income, complicating individuals' ability to afford food. Remittances from expatriate workers, an essential economic source for numerous African families, decreased as global economies contracted. The economic recession induced by the epidemic exacerbated poverty, hence restricting individuals' access to food. The pandemic resulted in heightened demand for specific food items, especially basics, leading to an increase in prices”
- Insufficient Empirical Methods
The paper employs cross-sectional data and logistic regression models, but the description of the empirical methods is insufficient. The sampling strategy, variable operationalization, and potential selection biases are not well elaborated. The definition of key variables, such as "socioeconomic status" and "overcrowding," lacks detail, and this lack of clarity can lead to misinterpretation of results. The measurement tools and methods of categorization are not clearly defined, which reduces the transparency of the methodology.
Suggestions: The methodology section should provide a detailed explanation of the sampling process, including how the surveyed areas were selected and how representativeness was ensured. A clearer explanation of how key variables were operationalized is essential. For instance, the criteria for classifying "socioeconomic quintiles" and determining "overcrowding" need to be defined clearly, possibly referencing widely accepted standards. This will enhance the robustness and replicability of the study. Furthermore, the analysis could benefit from a discussion of the potential biases that could arise from the sampling or data collection methods, such as selection bias or response bias, and how the study design addresses or mitigates these concerns.
We have addressed this in the revised version, which can be seen as follows: The details about sample size requirements are provided in the manuscript. This study used a two-stage cluster design. A representative sample of enumeration areas (EAs) were drawn from a master sampling frame covered, provided by the national statistical agency. Ahead of data collection, households, and key landmarks in each EA were listed and mapped by data collectors. Within each EA, a random sample of households were selected. The survey aims to include a sample size that would allow analysts to calculate an estimate at Kinshasa and by slum settlements, with a margin of error of ±3 percentage points.
The household wealth index was constructed based on principal component analysis [25] to create an index from a set of household assets (radio, tape recorder, television set, bicycle, hand torch, horse or donkey cart), housing conditions (roof material, number of rooms, wall type, windows, availability and type of latrine), and ownership of domestic animals. The study participants were ranked according to the wealth index score, divided into quintiles, from the lowest (first quintile) to the highest (fifth quintile). Overcrowded households were defined as those with four or more persons living in one room.
Discussion of the potential biases was addressed in discussion section.
- Logical Issues in Argumentation
The logical flow of the paper is inconsistent, particularly between the results and recommendations. The authors identify several factors associated with food insecurity, such as age, overcrowding, and socioeconomic status, but the link between these factors and the policy recommendations is weak. For instance, the paper suggests interventions like cash transfers and food assistance programs, but these are not clearly connected to the empirical findings.
Suggestions: The paper needs a more cohesive structure, with clear logical connections between the empirical results and the recommendations. For example, if overcrowding is a key factor linked to food insecurity, the recommendations should specifically address how improving housing conditions could help alleviate food insecurity. Similarly, interventions like cash transfers should be explicitly tied to the analysis of income or socioeconomic data. Each recommendation should flow directly from the study’s findings, ensuring the argument is clear and well-supported throughout the paper.
- Weak Connection Between Results and Recommendations
While the paper identifies key predictors of food insecurity, such as poverty, overcrowding, and slum residence, the proposed policy recommendations seem generic and not directly derived from the results. For instance, the suggestion of cash transfers lacks a clear explanation of how it would address the specific causes of food insecurity identified in the study, such as poor housing conditions or informal employment. The disconnect between findings and suggestions reduces the practical relevance of the recommendations.
Suggestions: Each policy recommendation should be directly tied to the study’s specific findings. For example, if socioeconomic status is a significant predictor of food insecurity, the recommendations should focus on targeted poverty alleviation strategies, such as income support or employment programs that directly address income disparities. Similarly, if overcrowding is a key issue, policies aimed at improving urban infrastructure and housing conditions should be considered. The recommendations should provide actionable strategies that directly address the factors identified in the empirical analysis.
Authors response: We have addressed this in the revised version, which can be seen as follows: “Through the implementation of specific interventions, households can markedly diminish food insecurity, enhance livelihoods, and attain sustainable agricultural development. Assist small-scale farmers by providing access to contemporary agricultural technologies, education, financing, and market opportunities. Formulate policies that endorse sustainable agriculture, enhance food security, and stimulate investment. Ensure secure land tenure rights for farmers to promote investment and enhance productivity. Facilitate the transmission of agricultural technologies from wealthy nations to African countries.”
- Incomplete Discussion of Study Limitations
Although the paper acknowledges some limitations, such as the lack of pre-pandemic data, the discussion is not comprehensive. It overlooks several important factors that could have influenced the results, such as differences in wealth distribution, education levels, and access to services between slum and non-slum areas. Additionally, the study relies heavily on self-reported data, which is prone to recall and reporting bias, yet this issue is not adequately addressed.
Suggestions: A more thorough discussion of the study’s limitations is needed. The authors should acknowledge the limitations of using self-reported data, such as recall bias, and discuss how this could have impacted the results. Furthermore, the limitations of the cross-sectional study design should be addressed, as this approach only allows for the identification of associations and not causality. The authors could suggest using longitudinal data or mixed-method approaches in future research to strengthen causal inferences and address the limitations of the current study design.
Authors response: We have addressed this in the revised version, which can be seen as follows: “We did not provide a qualitative description of the food system of households in Kinshasa, including its primary characteristics, food typologies, and dishes in both slum and non-slum areas. In this study, evaluation of confounding was particularly relevant in the identification of factors associated with food insecurity. Most of the confounders were in fact controlled during the data analysis stage, using multivariate analyses. Even if we had used a multivariate technique, there can remain residual confounding in this study. There was the probability that the additional confounding factors were not considered, because data on these factors were not collected. Self-reported data and recall bias must be considered when interpreting the study results.”
- Lack of Gender Analysis
Despite focusing on household food insecurity, the paper does not explore gender differences, which are often significant in studies of food insecurity. Women and children, especially in urban poor settings, are more likely to experience food insecurity, yet the paper does not investigate whether female-headed households face higher risks or how gender roles within households may impact food distribution.
Suggestions: Incorporating gender as a key variable could provide valuable insights into the dynamics of food insecurity in slums and non-slums. The authors should analyze whether female-headed households are more vulnerable to food insecurity and examine the role of gender in household food distribution. This could also lead to more targeted recommendations, such as empowering women through livelihood programs or increasing access to social services for female-headed households. Including a gender analysis would strengthen the study and contribute to a more nuanced understanding of the factors influencing food insecurity.
Authors response: Thanks for your comment and we find it valuable. The tool we used to measure Food security is measuring at household level. However, in the regression model, we used the sex of the head of the Household.
- Lack of Specificity in Policy Recommendations
The paper’s policy recommendations are broad and lack concrete implementation strategies. For example, while cash transfers and food assistance are proposed as solutions, the recommendations do not include details about how these programs would be rolled out, who would be targeted, or how effectiveness would be monitored. Without these specifics, the recommendations appear less practical and actionable.
Suggestions: The recommendations should be more specific and actionable. For instance, the paper could outline how cash transfer programs could be implemented in slum areas, with clear criteria for identifying beneficiaries and mechanisms for delivering aid. The authors could also discuss potential challenges, such as administrative capacity or targeting efficiency, and suggest ways to address these. Drawing on examples from other countries or regions where similar interventions have been successfully implemented would provide empirical support for the recommendations and make them more credible.
Authors response: Thanks for your comment and we find it valuable. This statement can address this comment: “Additionally, to combat economic restrictions that lead to food insecurity during crises, policymakers and implementing partners might enhance food assistance programs, such as cash transfers and food supply initiatives, focusing on overcrowded households and the informal job sector.”
- Limited Cross-Sectoral Discussion
Although the paper discusses food insecurity, it does not sufficiently address the broader contextual factors that contribute to this issue, such as infrastructure, access to clean water, sanitation, and public health services. These factors are crucial in understanding the full scope of food insecurity, particularly in urban slums where overcrowding and poor living conditions exacerbate the problem.
Suggestions: The authors should expand the scope of the discussion to include how cross-sectoral factors such as urban infrastructure, public health, and sanitation affect food security. For instance, they could explore the link between poor sanitation and food utilization or how lack of access to clean water in slums contributes to poor nutritional outcomes. By addressing these broader factors, the paper would provide a more comprehensive understanding of the challenges of food insecurity and offer more holistic policy solutions that tackle the issue from multiple angles.
Authors response: This was addressed as one of the weaknesses of this study. This statement can address this comment: “This study did not assess the link between poor sanitation and food utilization or how lack of access to clean water in slums contributes to poor nutritional outcomes.”
Comments on the Quality of English Language
Several grammar issues have been identified that impact the overall clarity and readability of the paper. Below is a comprehensive explanation of these issues and recommendations for improvement.
One of the most noticeable grammar problems in the paper is the inconsistency in verb tense. The paper frequently shifts between past, present, and future tenses, often within the same paragraph or sentence. This inconsistency makes it difficult for the reader to follow the timeline of the research or understand whether the statements refer to past studies, present circumstances, or future implications. For instance, when discussing research that has already been conducted, the past tense should be consistently used, while general statements of fact can be written in the present tense. Maintaining tense consistency throughout the paper will create a more coherent and professional narrative.
Another issue is subject-verb agreement. In many cases, plural subjects are mistakenly paired with singular verbs, and vice versa. This type of error can confuse the reader and detract from the academic tone of the paper. For example, in sentences where the subject is plural, such as "the data from the surveys," the verb must also be plural to match, as in "the data from the surveys show." Ensuring that subjects and verbs agree in number throughout the paper is essential for clarity and correctness.
Additionally, the paper contains both sentence fragments and run-on sentences, which can affect the flow of ideas and make the text harder to comprehend. Sentence fragments occur when incomplete thoughts are presented without a main clause, while run-on sentences happen when multiple independent clauses are joined without proper punctuation or conjunctions. To resolve these issues, it is important to structure sentences so that each expresses a complete idea and uses appropriate punctuation to separate or connect thoughts. For instance, joining two independent clauses with a conjunction like “and” or “but” or using a semicolon to link closely related ideas can improve the overall sentence structure.
Redundancy and wordiness also appear frequently in the text, where ideas are unnecessarily repeated or expressed in a verbose manner. This diminishes the paper's clarity and can cause the writing to feel repetitive or overly complicated. To address this, aim to express ideas more concisely and avoid restating the same concept multiple times within a single section. For example, instead of stating, "It is very clear and evident that there is a strong and significant relationship between income level and food insecurity," a more concise version would be, "There is a significant relationship between income level and food insecurity."
The misuse of articles (a, an, the) is another common issue found in the document. The incorrect or inconsistent use of articles can alter the meaning of a sentence or make it grammatically incorrect. For instance, phrases like "due to lack of proper housing" should be corrected to "due to **a** lack of proper housing." When referring to specific or previously mentioned nouns, the definite article "the" should be used, whereas indefinite articles "a" or "an" should be used for more general or singular references.
Another issue involves unclear or ambiguous pronoun references. In several instances, pronouns like “it” or “they” do not clearly refer to a specific noun, which can confuse the reader about what or who is being discussed. To improve clarity, make sure that every pronoun has a clear antecedent. When necessary, it’s better to restate the noun to avoid confusion, particularly when several potential antecedents are present in the same sentence.
There are also instances of incorrect preposition use throughout the text. Prepositions such as "on," "at," "in," and "by" must be used correctly to maintain grammatical accuracy and convey the intended meaning. For example, phrases like "aimed on improving food security" should be corrected to "aimed **at** improving food security." Ensuring proper preposition use throughout the paper will help eliminate awkward phrasing and make the arguments more precise.
The overuse of passive voice is another concern. While the passive voice is acceptable in academic writing, an excessive use of it can make the text less direct and harder to follow. In many cases, using the active voice can make sentences clearer and more engaging. For example, instead of saying “It was found that many households were food insecure,” the active voice could be used to say “The survey found that many households were food insecure.” This switch to the active voice emphasizes the subject and clarifies who or what is performing the action.
Another problem identified is the presence of comma splices, where two independent clauses are incorrectly joined by a comma without the proper use of a coordinating conjunction. Comma splices create confusion and detract from the sentence’s readability. For instance, a sentence like “The economy in Kinshasa is struggling, food insecurity continues to rise” should be corrected by adding a conjunction: “The economy in Kinshasa is struggling, and food insecurity continues to rise.” Alternatively, a period could be used to split the clauses into two distinct sentences.
Finally, some instances of incorrect word choice were identified in the text. Misusing certain words can distort the meaning of a sentence or make it sound awkward. For example, using the phrase "detrimental to" instead of "negatively impacts" may not be the most appropriate choice in certain contexts. Choosing more precise words will enhance the clarity and academic tone of the writing. It’s also important to use vocabulary that accurately conveys the intended meaning, avoiding unnecessary complexity or ambiguity.
By addressing these grammatical issues—such as verb tense consistency, subject-verb agreement, sentence structure, redundancy, article use, pronoun clarity, preposition choice, voice (active vs. passive), comma splices, and word choice—the overall quality, coherence, and professionalism of the paper can be significantly improved. Paying attention to these details will not only enhance readability but also ensure that the arguments and findings are communicated clearly and effectively.
Authors response: Thanks for these comments, before submitting the manuscript. We have submitted the first version to MDPI English editing service, and We are open to request further edits from the same services

Round 2
Reviewer 3 Report
Comments and Suggestions for Authors
The authors have made commendable efforts to address my previous feedback by enriching the content. However, the revisions do not fully meet the expectations I have for a manuscript of this caliber. The logical flow and alignment between sections, especially the theoretical foundation and how it connects with subsequent analysis and findings, require further refinement. Specifically, the paper would benefit from a more cohesive integration of theoretical concepts throughout, ensuring each section logically builds upon the last. Strengthening this aspect would improve clarity and offer a more compelling narrative that aligns the theory, methodology, results, and implications into a unified argument.
Comments on the Quality of English LanguageFurther improvement in both grammar and sentence structure is necessary.
Author Response
Second round of reviews:
The authors have made commendable efforts to address my previous feedback by enriching the content. However, the revisions do not fully meet the expectations I have for a manuscript of this caliber. The logical flow and alignment between sections, especially the theoretical foundation and how it connects with subsequent analysis and findings, require further refinement. Specifically, the paper would benefit from a more cohesive integration of theoretical concepts throughout, ensuring each section logically builds upon the last. Strengthening this aspect would improve clarity and offer a more compelling narrative that aligns the theory, methodology, results, and implications into a unified argument.
Authors’ reply:
Dear Reviewers,
Many thanks for your thoughtful comments.
We have revised the manuscript to address your concerns, particularly regarding the integration and alignment of the theoretical foundation with subsequent sections.
Strengthening theoretical framework integration: we have refined the theoretical framework at the outset and ensured that it guides the rest of the manuscript, especially the discussion. Specifically, we aligned our focus on food security dimensions—availability, access, vulnerability, and utilization—with the methodology, data analysis, and discussion.
Enhanced logical flow: to improve coherence, we tried to establish clearer connections between the conceptual framework and our findings, highlighting how specific factors—such as socioeconomic status, informal employment, and living conditions in slum vs. non-slum areas—contribute to food insecurity, particularly during crises like the COVID-19 pandemic.
Unified narrative and cohesion: we ensured each section builds logically upon the previous one, linking the theoretical framework to the results, and providing policy recommendations grounded in our findings. We believe this clarifies our argument and emphasizes the practical implications of our study.
We hope these improvements fulfill your expectations, and we thank you for helping us enhance the quality and coherence of our work.
